# Proximal Pulmonary Fat Embolism on Non-Contrast Chest CT

**DOI:** 10.3390/diagnostics15192468

**Published:** 2025-09-26

**Authors:** Romain L’Huillier, Alexandra Braillon

**Affiliations:** 1Department of Medical Imaging, Edouard Herriot Hospital, Hospices Civils de Lyon, University of Lyon, 69002 Lyon, France; 2LabTAU, INSERM U1032, 69003 Lyon, France; 3Everest, The French Comprehensive Liver Center, Hospices Civils de Lyon, University of Lyon, 69002 Lyon, France; 4Department of Medical Imaging, Louis Pradel Hospital, Hospices Civils de Lyon, 69002 Lyon, France; alexandra.braillon01@chu-lyon.fr

**Keywords:** pulmonary fat embolism, fat embolism syndrome, nonthrombotic pulmonary embolism

## Abstract

We report in this clinical case a proximal pulmonary fat embolism detected on unenhanced chest computed tomography (CT) responsible for a recovered cardiac arrest during a left total hip arthroplasty for a femoral neck fracture. This observation underscores the diagnostic value of integrating a non-contrast phase in chest CT in the postoperative context of orthopedic surgery, as it ensures accurate identification of the fatty nature of pulmonary arterial thrombi and thereby contributes to improved diagnostic accuracy and differential diagnosis.

**Figure 1 diagnostics-15-02468-f001:**
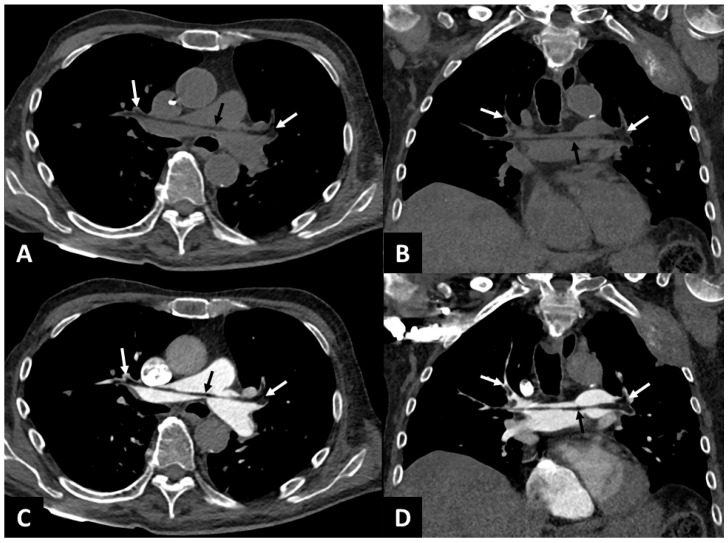
An 85-year-old patient with a medical history of rectal adenocarcinoma with hepatic and osseous metastases presented with a non-displaced secondary fracture of the left femoral neck. The patient underwent left total hip arthroplasty, during which an intraoperative cardiac arrest occurred. Return of spontaneous circulation was successfully achieved with epinephrine administration. Conventional chest CT (Monotube SOMATOM^®^ Definition Edge, Siemens Healthineers, Erlangen, Germany) without and with intravenous injection of iodinated contrast medium (60 cc, 3 cc/s) in the pulmonary arterial phase was performed 20 min after the cardiac arrest. The technical parameters were as follows: slice thickness 1 mm, 100 kVp, and 300 mAs on average (modulation in *z* axis). Images were reconstructed using a standard filtered backprojection kernel (Br38), and the windowing used was WL 50 HU and WW 350 HU. Unenhanced chest CT in axial (**A**) and coronal (**B**) reconstructions revealed negative density content in the pulmonary artery (black arrow, −50 HU with ROI of 5 mm^2^), in the right (−84 HU with ROI of 10 mm^2^) and left branches (−97 HU with ROI of 21 mm^2^) of the pulmonary artery, in the right upper lobar branch (−95 HU with ROI of 8 mm^2^), and the left upper lobar branch (−92 HU with ROI of 14 mm^2^) (white arrows). CT images in the axial plane (**C**) and coronal plane (**D**) in the pulmonary arterial phase revealed a pulmonary artery enhancement defect (in place of the negative density content visible on the non-contrast CT) in the pulmonary artery (black arrow), in the right and left branches of the pulmonary artery, and in the right and left upper lobar branches (white arrows). The negative attenuation on the non-contrast study is less visible on a computed tomography pulmonary angiogram (CTPA) because iodine admixture leads to a relative isoattenuation and fat thrombi appear only as filling defects. The patient responded favorably to anticoagulant therapy (introduced because of the initial cardiac arrest) and oxygen therapy was discontinued after 2 days without secondary acute respiratory failure. The early recognition of pulmonary fat embolism enabled rapid adjustment of preoperative management, notably by instituting close monitoring for clinical signs consistent with fat embolism syndrome (FES) potentially associated with this proximal pulmonary fat embolism. FES is a well-recognized complication of long bone fractures and orthopedic surgeries [1], characterized by pulmonary (hypoxia), neurological, and cutaneous manifestations [2] resulting from the circulation of fat droplets within the systemic and pulmonary capillaries [3]. Thoracic computed tomography is a key component in the diagnosis, and the most common findings are ground-glass opacities and air-space consolidations related to non-specific alveolar damage due to distal obstruction of pulmonary circulation [4]. The visualization of proximal fat thrombi in the pulmonary arterial circulation is exceptional [5] and made difficult by the use of a CT scan with intravenous contrast agent injection [6]. In addition to detecting fat emboli, unenhanced CT can also help identify other causes of nonthrombotic pulmonary embolism by detecting low attenuation intravascular content (as is observed with gas (−1000 HU) in cases of air embolism) or material with high attenuation (cement, lipiodol) [7]. In our case, the diagnosis of fat embolism was deemed most probable, taking into account the clinical context of recent orthopedic surgery. Dual-energy CT has been evaluated in rabbits for the detection of distal thrombi in fat embolism syndrome [8] and could be useful for the detection of proximal fat thrombi, despite the injection of iodinated contrast medium, thanks to virtually non-contrast reconstructions. This observation shows the possible visualization of fat thrombi in the pulmonary circulation on unenhanced CT in cases of pulmonary fat embolism syndrome. In postoperative scenarios, the use of an unenhanced phase can increase the detection of fat thrombi because of the intrinsic contrast between fat (negative attenuation) and blood that the iodinated contrast may mask, converting the appearance to a filling defect.

## Data Availability

The data presented in this study is available on request from the corresponding author. The data is not publicly available as it contains confidential doctor and patient information.

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
