# Peer review of "Proximal Pulmonary Fat Embolism on Non-Contrast Chest CT"

_diagnostics, 2025, doi:10.3390/diagnostics15192468_

Round 1
Reviewer 1 Report
Comments and Suggestions for Authors
An 85‑year‑old patient experienced intraoperative cardiac arrest during left total hip arthroplasty. The noncontrast chest CT demonstrates intraluminal material with negative attenuation (~−100 HU) in the main pulmonary artery and right/left main branches; the subsequent CT pulmonary angiography (CTPA) shows corresponding filling defects. Clinical course was favorable with oxygen therapy and anticoagulation. The images (Figure 1, panels A–D) are clear and consistent with the proposed interpretation.
Overall Assessment
This is educationally strong and clinically relevant. It underscores that intravascular fat can be directly detected on unenhanced CT, which is uncommon compared with the better‑known parenchymal manifestations of fat embolism syndrome (FES). The “Interesting Images” format fits the content; the narrative is concise and the iconography is good.
The strengths of this article are
- Clear take‑home message: intraluminal fat may appear as negative attenuation on noncontrast CT and later as a conventional filling defect on CTPA.
- Clinical relevance: endoluminal fat visualization in FES is rare; highlighting it is helpful for emergency workflows.
- References are compact yet current, spanning foundational and recent literature, including 2024–2025 items.
Other suggestions are recommended
- Densitometry and ROI methodology. Specify how HU were measured: ROI size and placement, reconstruction kernel, slice thickness, and window/level; address partial‑volume avoidance. Report HU values for each site where fat is claimed (trunk, main, and lobar branches). If feasible, add an inset with ROI screenshots (panels A/B).
- Therapeutic framing. The text credits improvement in part to anticoagulation. For macroscopic fat embolism, contemporary literature emphasizes supportive care; anticoagulation is not uniformly indicated unless coexistent thrombus is suspected. Please justify the anticoagulation (e.g., clinical/lab grounds, echocardiography, D‑dimer, subsequent CTPA) and tone down any causal implication.
- Differential diagnosis with air and other nonthrombotic emboli. Add a brief paragraph distinguishing air (≈−1000 HU, nondependent morphology), lipiodol, tumor or cement emboli (typically high attenuation), and place this case within the broader category of nonthrombotic pulmonary emboli.
- Terminology consistency. Standardize fat embolism syndrome (FES) (systemic clinical entity) versus pulmonary fat embolism (vascular event). If you cite historical clinical criteria (e.g., Gurd), do so succinctly.
- Imaging technique context. Provide technical parameters: detector configuration, slice thickness, kernel, kVp/mAs, timing relative to the event, and contrast volume/rate for CTPA. This improves reproducibility in acute care settings.
- Clarify the “noncontrast‑first” message. State explicitly that, in postoperative scenarios, an unenhanced phase can increase intrinsic contrast between fat and blood; iodinated contrast may mask negative attenuation, converting the appearance to a filling defect. A one‑sentence note on the potential role of dual‑energy CT would strengthen this point.
In addition
- Units: correct “−100 UH” → “−100 HU (Hounsfield units)”.
- Typos: “lef upper lobar” → “left upper lobar”; “WriĴen” → “Written”. Remove spurious hyphenation breaks (“pulmo‑nary,” etc.).
- English variety: choose American English consistently (e.g., “favorable”) or clearly adhere to the journal’s preferred style and apply it uniformly.
- Keywords: standardize punctuation, e.g., “pulmonary fat embolism; fat embolism syndrome; nonthrombotic pulmonary embolism.”
- Figure caption: report window/level used for HU measurements; ensure arrow/arrowhead conventions are consistent; add scale and orientation markers if possible.
- Ethics: For a single case, patient consent is usually sufficient; you report both IRB approval and written consent, which is fine. Just make sure wording matches the journal’s policy language.
References — Verification and Adequacy
Six citations are listed; all appear valid and appropriate:
- Akhtar, 2009 (Anesthesiology Clinics): authoritative review on FES.
- Gurd, 1970 (JBJS Br): historical clinical criteria; acceptable as a classic.
- Kosova et al., 2015 (Circulation): succinct, high‑quality overview.
- Newbigin et al., 2016 (Respiratory Medicine): imaging‑focused review; central to your message.
- Brun & Ghaye, 2025 (Revue des Maladies Respiratoires): up‑to‑date review on nonthrombotic emboli; highly pertinent.
- Murphy et al.,2024 (Radiology Case Reports): macroscopic pulmonary fat embolism on CTPA; emphasizes avoiding unnecessary anticoagulation.
Optional but useful additions:
- A broad imaging review of nonthrombotic pulmonary artery emboli (e.g., AJR, ~2017) to anchor the differential.
- A brief DECT reference related to FES (e.g., European Radiology, ~2017, experimental/animal model) to support the technical note.
Bottom line on references: concise yet qualified and current; adding 1–2 targeted items would complete the conceptual frame.
Targeted Suggestions for Figure 1 and for the Text
- Figure 1: add a magnified inset with ROI overlays and mean/SD HU in at least two locations; state slice thickness (e.g., 1–2 mm) and reconstruction kernel. Consider an additional panel with MIP or sagittal MPR to depict distribution.
- Main text (around the description of A–D): provide numeric HU values and explicitly explain why the negative attenuation on the noncontrast study is not visible on CTPA (iodine admixture → relative isoattenuation and presentation as a filling defect).
- Outcome details: specify timing (hours/days) between the event, the noncontrast CT, and CTPA; include ABG parameters and D‑dimer if available.
Author Response
Thank you for your valuable review.
All comments have been taken into account. The changes are highlighted in yellow in the manuscript.
1 - Densitometry and ROI methodology. Specify how HU were measured: ROI size and placement, reconstruction kernel, slice thickness, and window/level; address partial‑volume avoidance. Report HU values for each site where fat is claimed (trunk, main, and lobar branches). If feasible, add an inset with ROI screenshots (panels A/B).
Trunk : ROI size 5 mm², -50 HU
Right branch : ROI size : 10mm², -84 HU
Left branch : ROI size 21 mm², -97 HU
Right superior lobar branch : ROI size 8mm² , -95 HU
Left superior lobar branch : ROI size 14 mm², -92 HU
Slice thickness : 1 mm
Window - level : 350 – 50
Reconstruction kernel : standard FBP kernel (Br38)).
2 - Therapeutic framing. The text credits improvement in part to anticoagulation. For macroscopic fat embolism, contemporary literature emphasizes supportive care; anticoagulation is not uniformly indicated unless coexistent thrombus is suspected. Please justify the anticoagulation (e.g., clinical/lab grounds, echocardiography, D‑dimer, subsequent CTPA) and tone down any causal implication.
The anticoagulation was introduced because of cardiac arrest and no thrombectomy was performed because of multidisciplinary discussion for limitation. D-dimers not available because of cardiac arrest and performing CTPA immediately after it.
3- Differential diagnosis with air and other nonthrombotic emboli. Add a brief paragraph distinguishing air (≈−1000 HU, nondependent morphology), lipiodol, tumor or cement emboli (typically high attenuation), and place this case within the broader category of nonthrombotic pulmonary emboli.
Added with reference.
4 - Terminology consistency. Standardize fat embolism syndrome (FES) (systemic clinical entity) versus pulmonary fat embolism (vascular event). If you cite historical clinical criteria (e.g., Gurd), do so succinctly.
Added.
5 - Imaging technique context. Provide technical parameters: detector configuration, slice thickness, kernel, kVp/mAs, timing relative to the event, and contrast volume/rate for CTPA. This improves reproducibility in acute care settings.
Monotube SIEMENS Somatom Definition Edge
Slice thickness : 1 mm
Kernel : Br38
kVp: 100 ; mAs: 300 (with z axis modulation)
60 cc of contrast medium at 3 cc/sec,
Time relative to event: 20 minutes after cardiac arrest
6 - Clarify the “noncontrast‑first” message. State explicitly that, in postoperative scenarios, an unenhanced phase can increase intrinsic contrast between fat and blood; iodinated contrast may mask negative attenuation, converting the appearance to a filling defect. A one‑sentence note on the potential role of dual‑energy CT would strengthen this point.
Added with reference.
Reviewer 2 Report
Comments and Suggestions for Authors
This is an impressive image demonstrating fat embolism in the pulmonary artery on non-contrast CT, and I believe many readers will be interested in it.
The authors also presented attenuation values of the emboli.
If the reference limit allows, it would be helpful to mention that pulmonary artery thrombi can sometimes be detected on non-contrast CT as well; for example, Hyperdense pulmonary artery sign - detection of pulmonary embolism in patients with suspected COVID-19 using non-contrast chest CT. Reinert D, et al. Radiol Case Rep. 2021.
Author Response
Merci pour votre revue précieuse.
Tous les commentaires ont été pris en compte. Les modifications sont surlignées en jaune dans le manuscrit.